# CXCL4-RNA Complexes Circulate in Systemic Sclerosis and Amplify Inflammatory/Pro-Fibrotic Responses by Myeloid Dendritic Cells

**DOI:** 10.3390/ijms24010653

**Published:** 2022-12-30

**Authors:** Immacolata Pietraforte, Alessia Butera, Lucia Gaddini, Anna Mennella, Raffaella Palazzo, Doriana Campanile, Katia Stefanantoni, Valeria Riccieri, Roberto Lande, Loredana Frasca

**Affiliations:** 1Istituto Superiore di Sanità, Department of Oncology and Molecular Medicine, 00161 Rome, Italy; 2Istituto Superiore di Sanità, National Center for Drug Research and Evaluation, 00161 Rome, Italy; 3Department of Clinical, Internal, Anesthesiological and Cardiovascular Sciences, Sapienza University of Rome, 00185 Roma, Italy

**Keywords:** CXCL4, RNA, systemic sclerosis, inflammation, myeloid dendritic cells (mDCs)

## Abstract

CXCL4 is an important biomarker of systemic sclerosis (SSc), an incurable autoimmune disease characterized by vasculopathy and skin/internal organs fibrosis. CXCL4 contributes to the type I interferon (IFN-I) signature, typical of at least half of SSc patients, and its presence is linked to an unfavorable prognosis. The mechanism implicated is CXCL4 binding to self-DNA, with the formation of complexes amplifying TLR9 stimulation in plasmacytoid dendritic cells (pDCs). Here, we demonstrate that, upon binding to self-RNA, CXCL4 protects the RNA from enzymatic degradation. As a consequence, CXCL4-RNA complexes persist in vivo. Indeed, we show for the first time that CXCL4-RNA complexes circulate in SSc plasma and correlate with both IFN-I and TNF-α. By using monocyte-derived DCs (MDDCs) pretreated with IFN-α as a model system (to mimic the SSc milieu of the IFN-I signature), we demonstrate that CXCL4-RNA complexes induce MDDC maturation and increase, in particular, pro-inflammatory TNF-α as well as IL-12, IL-23, IL-8, and pro-collagen, mainly in a TLR7/8-dependent but CXCR3-independent manner. In contrast, MDDCs produced IL-6 and fibronectin independently in their CXCL4 RNA-binding ability. These findings support a role for CXCL4-RNA complexes, besides CXCL4-DNA complexes, in immune amplification via the modulation of myeloid DC effector functions in SSc and also during normal immune responses.

## 1. Introduction

CXCL4 is an important biomarker of systemic sclerosis, an autoimmune disease characterized by vasculopathy, inflammation, and autoimmunity [1,2]. These three hallmarks lead to the fibrosis of the skin (limited SSc form) and internal organs (diffuse form). The disease is characterized by a type I interferon (IFN-I) signature in about half of the patients, which may have several deleterious effects, and predicts worse disease progression when present at disease onset [3,4,5]. It is important to point out that IFN-I plays many functions in immune activation and primes immune cells towards a higher response to RNA ligands [3,4,5]. We have previously demonstrated that CXCL4 binds self/microbial DNA and forms nanocrystalline complexes that optimally activate TLR9 in plasmacytoid DC (pDC), leading to the amplification of IFN-I production in response to the DNA [6]. In addition, when CXCL4 was mixed with RNA and administered to pDC, a certain amount of IFN-α was also produced [7]. Similarly, CXCL4 and RNA activated memory B-cells pretreated with IFN-α [8]. Here, we show that, after binding self-RNA, CXCL4 preserves the bound RNA from degradation, a phenomenon already observed for self-DNA [6]. Although CXCL4-DNA complexes were shown in the blood and skin of SSc patients [6], whether CXCL4-RNA complexes also persist in SSc is unknown. We found that such complexes exist in SSc plasma, which prompted us to address their role not only as stimulators of pDC but also of myeloid DC (mDC). Indeed, human mDCs express RNA-sensing receptors and are master regulators of immunity [9,10,11,12,13,14,15,16]. Signals that elicit important pro-inflammatory programs may favor hyperactivation and autoimmunity and drive pathways involved in fibrosis [17,18]. Differentiated monocyte-derived dendritic cells (MDDCs) can be derived from human blood monocytes, and these cells can be considered the in vitro counterpart of inflammatory monocytes that became DCs and homes to inflamed sites [10,11,17,18]. We thus used MDDCs as an in vitro model system to assess the immune amplification effect of CXCL4 and RNA on several cell factors involved in SSc pathogenesis and fibrosis. Our findings indicate that CXCL4 affects mDC activation, as previously demonstrated [17,18]. However, the CXCL4 effect also depends on its capacity to bind self-RNA, leading to immune amplification via endosomal RNA-sensing receptors. Thus, beyond affecting the IFN-I signature, CXCL4-RNA complexes can modulate other important immune cells, such as mDCs, and amplify pro-inflammatory factors other than IFN-I as well as pro-fibrotic mediators.

## 2. Results

### 2.1. CXCL4 Binds Self-RNA in a Dose-Dependent Manner and Prevents Its Degradation

We have previously demonstrated that CXCL4 premixed with RNA stimulates the production of IFN-I by pDC and activates B-cells pretreated with IFN-α [7,8]. Here, we visualize this binding by electrophoretic mobility shift assay (EMSA) [6]. Figure 1a shows how CXCL4 binds the RNA extracted from human PBMCs (self-RNA). Binding started to be evident at 0.5 μMolar concentration in the presence of 20 μg per ml of self-RNA and was optimal between 2 and 4 CXCL4 μMolar concentrations (Figure 1a). Figure 2b shows that at 2 μMolar concentration, CXCL4 successfully protected the bound RNA from degradation mediated by partial digestion, with gradual concentrations of a ribonuclease (RNAse). These results suggest that CXCL4 condenses the self-RNA and impedes its processing by RNAse, a mechanism similar to that observed with self-DNA, where the DNA was protected by nucleases [6]. This can explain how self-RNA is rendered immunogenic for pDCs, as previously shown, and provides a mechanism for its persistence in vivo.

### 2.2. CXCL4-RNA Complexes Are Present in SSc Plasma and Correlate with IFN-I

We have previously shown that CXCL4, which is upregulated in SSc compared to healthy donor (HD) plasma [19], also circulates in SSc blood in the form of complexes with DNA [6]. Self-RNA is released by cells dying via several pathways, together with self-DNA. Although self-RNA is delicate and is even more rapidly degraded by ribonucleases in vivo, the fact that CXCL4 protected the self-RNA in vitro with a certain efficiency, as in Figure 1b, prompted us to look for CXCL4-RNA complexes in SSc plasma. To do so, we used a modified version of an ELISA assay previously set up to detect CXCL4-DNA complexes in SSc plasma [6]. ELISA plates were coated with CXCL4, and the RNA attached to the captured CXCL4 was revealed by using an anti-human RNA antibody. We first verified that our SSc cohort (see Table 1 for the demographic data of patients and controls) expressed CXCL4 in plasma (Appendix A). Twenty-four out of thirty-four SSc patients (70%) expressed variable amounts of CXCL4 in their plasma. Next, we tested for the presence of CXCL4-RNA complexes in the twenty-four samples of CXCL4-positive plasma (Figure 2a). Twelve out of twenty-four patients (50%) showed positivity for circulating CXCL4-RNA complexes. When we plotted the ELISA data (as optical density values (OD)) against the plasma concentration of the IFN-α measured by commercial ELISA, we found a significant positive correlation (*r_spearman_ =* 0.47; *p* = 0.01; *N* = 24) (Figure 2b). This reinforced our findings that CXCL4-RNA complexes induce IFN-α release by pDCs [7]. After selecting only the SSc plasma samples that were double-positive for CXCL4 and RNA (therefore containing CXCL4-RNA complexes), we calculated the correlation again with IFN-α and found that the coefficient « r » increased (*r_spearman_* = 0.77; *p* = 0.005; *N* = 12, Appendix A). In order to visualize CXCL4-RNA complexes, we selected SSc plasma containing these complexes (Figure 2c) and performed immunoprecipitation using an anti-CXCL4 antibody. When we ran the immune-precipitated material on Tricine gel, we were able to detect RNA co-immunoprecipitated with CXCL4 (Figure 2c). The RNA gave a signal on the gel where the tetrameric or dimeric forms of CXCL4 are usually detected. These results suggest that CXCL4-RNA complexes really circulate in SSc plasma. Thus, they are likely to be relevant for stimulating RNA sensors inside pDCs and other cell types [7,8] in vivo.

### 2.3. CXCL4-RNA Complexes Increase the Maturation of IFN-I-Primed Myeloid DCs

Having established that CXCL4-RNA complexes are stable in vitro and in vivo, we went on to address whether mDC could also be affected by these complexes. Such cells express TLR8 and TLR3, and TLR7 if stimulated by IFN-I [13,14,15]. Since SSc mDCs in vivo are likely to be exposed to IFN-I by the effect of the IFN-I signature often detected in these patients [1,2,5,6,19], we pretreated the MDDCs with 1000 UI/mL of IFN-α and stimulated them with CXCL4-RNA complexes for 48 h. A stimulation with CXCL4 alone already induced a significant maturation of MDDCs, measured as an increase of CD80 and CD86 double-positive cells by flow cytometry (cumulative data in Figure 3a, and representative experiment in Figure 3b). A 2 μMolar concentration of CXCL4 bound to human RNA was sufficient to significantly implement the upregulation of the MDDC maturation markers CD80/86 [11,12], as compared to CXCL4 alone or RNA alone. Although the administration of CXCL4 is not a null event for MDDC activation [17,18], CXCL4-RNA complexes amplify maturation, thus increasing mDC activation. Figure 3b,c show how different concentrations of CXCL4, ranging between 1 and 4 μMolars, alone or in complex with RNA, induced MDDC maturation. Of note, the maturation of MDDCs that were not pretreated with IFN-α was not significantly amplified by CXCL4-RNA complexes, suggesting that priming with IFN-I is important for this effect (Appendix A).

### 2.4. CXCL4-RNA Complexes Stimulate Pro-Inflammatory and Pro-Fibrotic Pathways

The stimulation of TLR in mDC leads not only to maturation but also to the secretion of cytokines, which play a major role in the intensity and polarization of the immune response. These factors affect both the innate and adaptive immune responses, as well as non-immune cells [15,18,20]. One of the main factors involved in SSc is TNF-α, and serum levels of TNF-α are elevated in patients with SSc and favor the development of pulmonary fibrosis and pulmonary arterial hypertension [21,22]. Thus, we wondered about the effects of CXCL4-RNA complexes (or CXCL4 alone) on the MDDC secretion of TNF-α and other additional factors in vitro. We found that different amounts of CXCL4 alone, added to MDDCs, negligibly affected TNF-α release, whereas the addition of CXCL4-RNA complexes greatly amplified the TNF-α secretion (Figure 4a). CXCL4-RNA complexes were also able to significantly increase IL-12 and IL-23 secretion compared to CXCL4 alone (Figure 4b,c). CXCL4 alone had some effect on both IL-12 and IL-23 (depending on the dose) with respect to untreated cells. As shown in Figure 5, the treatment of MDDCs with CXCL4-RNA complexes significantly amplified the production of IL-8 (also known as CXCL8) (Figure 5a), a pro-inflammatory chemokine that can be implicated in fibrosis and is induced by TNF-α [23,24,25]. Moreover, the stimulation determined an increase in pro-collagen production [26], although this effect was mainly observed at the highest complex doses (Figure 5b). Being important in SSc inflammation and fibrosis, we assessed IL-6 and fibronectin production (Appendix A) [26,27,28,29]. MDDCs stimulated with CXCL4 alone or CXCL4-RNA-complexes released IL-6 and fibronectin in response to CXCL4 alone, with no further significant amplification of this production by CXCL4-RNA complexes (Appendix A). These results suggest that CXCL4 Induces cytokines/chemokine release by MDDCs, but that CXCL4-RNA complexes amplify CXCL4 effects, at least for some of the factors induced.

### 2.5. CXCL4-RNA Complex Stimulation Is RNA-Sensing–TLR-Dependent and Independent of CXCR3

As mentioned above, myeloid cells can express RNA-sensing receptors in the cytosol and the endosomes [13,14,15,30,31]. Among the TLRs expressed in MDDCs, TLR3 recognizes double-strand RNA (dsRNA), and TLR7 and TLR8 are specifically for single-strand RNA (ssRNA). MDDCs express RNA-sensing TLR3, TLR8, and TLR7, depending on the treatment used for their differentiation. Upon IFN-I treatment, MDDCs express and even upregulate all TLR3, TLR7, and TLR8 and respond better to their specific ligands [14,15,16]. To address the way CXCL4-RNA complexes amplify cytokines, we treated MDDCs with bafilomycin, which is an inhibitor of activation of nucleic-acid-sensing endosomal receptors. The addition of bafilomycin (200 mM, Figure 6a) inhibited the production of TNF-α (Figure 6a) and also that of IL-12 and IL-23 (Appendix A). We also treated the MDDCs with two specific inhibitors of RNA-sensing TLRs (a TLR8 and a TLR7/8 inhibitor), as shown in Figure 6b. Both inhibitors significantly decreased the production of TNF-α by the cells. This suggested that the engagement of RNA-sensing TLRs was responsible for the observed effect. In contrast, the use of a blocking antibody to CXCR3, the receptor of CXCL4 [32], did not affect cytokine production by MDDCs during stimulation (Figure 6c and Appendix A), suggesting that the main CXCL4 receptor, CXCR3, is dispensable for MDDC activation by CXCL4-RNA complexes.

### 2.6. CXCL4-RNA Complexes Correlate with TNF-α in Plasma

The results presented above suggest that CXCL4-RNA complexes concur to mature and activate pro-inflammatory factors’ secretion of mDC. In our hands, TNF-α seemed particularly amplified by CXCL4-RNA complexes via RNA-sensing TLRs. Thus, we wondered whether CXCL4-RNA complexes, determined by ELISA in SSc plasma, could correlate with the TNF-α measured in the same plasma. SSc patients had a low but significant TNF-α upregulation in circulation (Figure 7a) compared to HDs. The intensity of the ELISA signal for CXCL4-RNA (OD) complexes correlated with TNF-α determined in the circulation in a moderate but significant manner (Figure 7b). When we restricted the correlation analysis to the patients that showed positivity for CXCL4-RNA complexes in plasma, the Spearman correlation coefficient « r » further increased, as well as the statistical significance of this correlation (Figure 7c). These results suggest that CXCL4-RNA complexes could concur with TNF-α production via myeloid cell stimulation in SSc. In contrast, we were unable to find a significant correlation between the concentration of IL-6 and the presence, in the same plasma, of CXCL4-RNA complexes (*r* = −0.14, *p* = 0.27, *N* = 22). These findings suggest that CXCL4-RNA complexes that are present in vivo could really contribute to TNF-α amplification.

## 3. Discussion

In this study, we have demonstrated for the first time that CXCL4-RNA complexes circulate in SSc blood and correlate with both IFN-I and TNF-α, two factors that are particularly upregulated in SSc and participate in disease pathogenesis [1,2,3,4,5,6,7,21,22,33].

These findings reinforce the role of CXCL4 and CXCL4-RNA complexes in the induction of the IFN-I signature in SSc, in addition to the role of CXCL4-DNA complexes [6], and extend the recognition of the effect of the CXCL4-RNA complexes in the activation of mDCs, which are crucial in immunity. CXCL4 and RNA have already been shown to concur with the activation of pDC and memory B-cells, as previously reported by us [7,8].

Here, we add important information about the capacity of CXCL4 to preserve RNA from degradation, which can explain the persistence in the circulation of self-RNA. This self-RNA contained in immune complexes is very likely to become immunogenic for immune cells such as pDCs and B-cells [7,8], as well as for mDCs.

Although the use of the MDDC model represents a simplification and a limitation, the data are interesting, as one of the main factors induced by CXCL4-RNA complexes in MDDCs in vitro was TNF-α [21,22,33]. This is in keeping with the evidence that TNF-α plasma levels correlated significantly with CXCL4-RNA complexes detected ex vivo. Moreover, we were able to show an effect on IL-12, IL-23, and IL-8 [34,35,36,37,38,39,40,41]. Interestingly, the same complexes were able to stimulate pro-collagen secretion, a factor involved in fibrosis [27,28,29]. Moreover, IL-8, which is also upregulated by CXCL4-RNA complexes, is a factor also involved in lung fibrosis [23,24]. IL-8 was found capable of expanding the mesenchymal cell population and recruiting activated macrophages to actively evolve fibrotic lesions [23,24].

These data have relevance not only in SSc but also for normal immune responses. Indeed, it is plausible that microbial RNA/DNA (either derived from viruses or bacteria) may exert similar stimulatory functions to mDC, in keeping with the antimicrobial activity of CXCL4, which is actually an antimicrobial chemokine [32]. CXCL4-RNA complexes were undetectable in HDs, but it is possible that in inflammatory conditions determined by pathogen invasion, microbial RNA (or DNA) binds to CXCL4, and the complexes formed stimulate RNA-sensing TLRs, potentiating an anti-infectious response.

Our data show that CXCL4, independent of its binding to RNA, also increased the secretion by MDDCs of fibronectin and IL-6. IL-6 production by MDDCs was thus independent of the complexes formed with RNA. The effect of CXCL4 alone, but not that of CXCL4-RNA complexes, has been studied previously on myeloid cells (monocytes and DCs) [17,18]. CXCL4 was given to MDDCs during their differentiation in vitro, dramatically changing their secretion pattern compared to cells that were not differentiated in the presence of CXCL4 [17,18]. Both blood DCs and classic MDDCs derived from SSc patients were found to respond in an increased manner to CXCL4, with the production of TNF-α, IL-6, and also the Th-2 type of cytokines. These effects were stronger upon TLR3 or TLR4 stimulation, with some increased responses also upon R848 stimulation (resiquimod), a TLR8 stimulator. As shown here, TNF-α was found to be upregulated in SSc patients’ plasma compared to HDs. In general, we have determined a lower level of plasma TNF-α in our cohort and also lower levels of IL-6 compared with previous work [35]. However, our data are in keeping with the finding that TNF-α and IL-6 can be upregulated in circulation in SSc compared to HDs. Interestingly, while TNF-α plasma concentrations correlated with CXCL4-RNA complexes in circulation, this was not observed with IL-6, in keeping with the finding that CXCL4-RNA complexes implement TNF-α but not IL-6 secretion by MDDCs in vitro.

The majority of the effects of CXCL4-RNA complex stimulation were observed after the pre-treatment of the MDDCs with IFN-α. This is not surprising as RNA-sensing TLRs are upregulated by IFN-I in MDDCs and in other cell types [13,14,15]. The complexes did not significantly affect untreated MDDCs. These findings further reinforce the important role of IFN-I in SSc [3,4,5,6]. Indeed, IFN-I not only has functions per se but also regulates the effector functions of several immune cells, in this case, the myeloid DC counterpart. Hence, the IFN-I signature in early SSc may not be the only parameter that is influenced by CXCL4 bound to nucleic acids. CXCL4 can stimulate some functions of the human myeloid DC [17,18] independently of its binding capacity to RNA; hence, CXCL4-RNA complexes do not necessarily need to form to have an increase in inflammation, determined by CXCL4. However, the immune amplification on mDC inflammatory functions via TLR7/8, after IFN-I priming, can represent a relevant additional contribution to SSc pathogenesis.

The data show that CXCR3, the CXCL4 receptor [32], is dispensable for the effect of CXCL4-RNA complexes on myeloid cells. The same phenomenon has been previously observed for human pDCs [6]. This means that CXCL4 bound to nucleic acid does not critically depend on the expression of the receptor CXCR3 to exert its function. It is not excluded that, at very low concentrations of such complexes, the role of the receptor CXCR3 may become more evident; however, at the concentration of CXCL4 used in this study and previous studies and given the very high expression of CXCL4 in SSc, the role of CXCR3 seems negligible [6,7,19].

The results reported here are interesting from the point of view of the pathogenesis of SSc. Indeed, TNF-α is a factor involved in SSc that favors pulmonary fibrosis and pulmonary arterial hypertension [25,34,36,37]. IL-23 is also elevated in SSc [34], and Th17-related cytokines (IL-17, IL-21, and IL-23) are associated with interstitial lung disease (ILD) and with its severity in SSc patients [34,35,36,38,39]. Serum levels of IL-12, increased in SSc patients, were described as associated with renal vascular damage [40,41].

The next step will be to analyze the expression of CXCL4-RNA complexes in various sub-groups (of sufficient size) of SSc patients presenting with different disease characteristics. Indeed, a limitation of the present study is that our SSc cohort was not so wide and was mostly homogeneous with respect to disease type. It comprised a very high number of patients with diffused cutaneous disease and ATA antibodies and only one patient with limited cutaneous SSc. It will be interesting to see whether the presence of CXCL4-RNA complexes can distinguish these two main forms of SSc as this information is missing in the study of actual cohorts. It is also possible that detectable circulating CXCL4-RNA complexes are markers of particular conditions or disease severity within the frame of the same disease. These questions are currently being investigated in our laboratory.

## 4. Materials and Methods

### 4.1. Patients

This study was approved by the ethical committee of the Sapienza University University, Rome, and conducted according to the Declaration of Helsinki. Blood samples (from 1 to 3 mL) from SSc patients were from Policlinico Umberto I, Rome, Italy. Plasma or sera from healthy donors (HDs), matched for age and sex with SSc patients as much as possible, were from the “blood centers”, Policlinico Umberto I, Italy. SSc patients satisfied the American College of Rheumatology (ACR)/European League Against Rheumatism (EULAR) 2013 classification criteria [42,43]. Exclusion criteria included patients treated with biologics. Plasma was obtained from whole blood collected in Vacutainer EDTA tubes (Becton and Dickinson, Franklin Lakes, NJ, USA) to avoid clotting. One milliliter of blood was centrifuged at 2000× *g* for 15 min. The supernatant was collected with a pipette and stored in 2 mL tubes at −80 °C for future experiments. Small aliquots of plasma were prepared to avoid freeze–thaw cycles.

### 4.2. Reagents and Antibodies

Antibodies for flow cytometry to CD80 and CD86, conjugated with fluorochromes (FITC, phycoerythrin (PE), or allophycocyanin (APC)), were from BD Biosciences or eBiosciences (San Diego, CA, USA). Appropriate isotype-matched controls were purchased from the same companies.

CXCL4 was from Novus Biological (Oxford, UK) or Biomatik (Roma, Italy). Bafilomycin A was from Sigma Aldrich (St. Louis, MI, USA) and was used at 50 nM. The TLR7/8 (Cat. No. CU-115, provided as a dried powder) and TLR8 (Cat. No CU-CPT9a, provided as a dried powder) inhibitors were purchased from MedChemExpress (Monmouth Junction, NJ, USA) and Invivogen (Toulouse, France), respectively, and used at concentrations of 50 and 40 µM.

### 4.3. Production of Human RNA

HuRNA was extracted from peripheral blood mononuclear cells (PBMCs) (from buffy coats, see below) by using the RNeasy Maxi Kit Qiagen, Cat. No. 75162, Dusseldorf, Germany). The resulting RNA was controlled by 2% agarose gel electrophoresis.

### 4.4. Binding of CXCL4 to RNA by Electrophoretic Mobility Shift (EMSA)

EMSA was performed by mixing various μM concentrations of CXCL4 with an appropriate concentration of RNA, which respected the appropriate protein–RNA ratio used in MDDC-stimulation assays. RNA alone and the mixtures were run on 2% agarose gel to evidence the delay in the migration of the RNA due to the binding to the tested proteins. The RNA on the gel was visualized with SYBR Safe DNA gel staining (ThermoFisher Scientific, Waltham, MA, USA).

### 4.5. Ribonuclease Protection Assay of RNA Bound to CXCL4

The suspension containing CXCL4–huRNA was incubated at 37 °C with different doses of DNase-free RNase A (28,000 units) from bovine pancreas, 100 U/µL (280 µL, Cat No. 26260, NORGEN, Thorold, ON, Canada) for 10–30 min at 37 °C. Then, the samples were run on agarose gel.

### 4.6. Isolation/Stimulation of Monocytes and Production of MDDCs

For the isolation of human peripheral blood monocytes, blood buffy coats of healthy donors (HDs) were obtained from the Blood Center of Policlinico Umberto I, Rome, IT. After the separation of PBMCs by Ficoll centrifugation, CD14^+^ monocytes were purified by using a CD14^+^ Cell Isolation Kit (Miltenyi Biotec, Bergisch-Gladbach, Germany). MDDCs were produced after culturing the monocytes with GM-CSF (Cat. No. 300-23, Preprotech, 100 µg provided as dried powder) and IL-4 (Cat. No. 200-04, Preprotech, 100 µg provided as dried powder) at 50 ng/mL for 5 days. After 5 days, the MDDCs harvested were treated with IFN-I (human recombinant IFN-α 4b, Cat. no. 11180-1, 100000 UN, and human recombinant IFN-α 2b, Cat. no. 11105-1, 100000 UN; R&D Systems, Minneapolis, MN, USA) at 1000 U/mL final concentration (500 U/mL 4b + 500 U/mL 2b). MDDC were seeded (the day after and after washing) into 96-well flat-bottom plates at 10^6^ cells per ml. Different amounts of CXCL4 were premixed with total human RNA (20 µg/mL) and added to the MDDC cultures after 15 min incubation at room temperature. Stimulation of MDDCs was also performed with CXCL4-RNA in the presence of the anti-human CXCR3 (R&D Systems, Minneapolis, MN, clone 49801 at 10 µg per ml) and control mouse IgG1 isotype control (10 µg per ml R&D). Antibodies were added to the MDDC cultures 30 min before adding the complexes to discriminate the pathway of activation. TLR inhibitors (see above) or bafilomycin A were added to the culture 15 min before adding the complexes of CXCL4-RNA. Cells were harvested after 48 h stimulation and analyzed for CD80 and CD86 upregulation (maturation). Culture supernantants, harvested after 48 h, were analyzed by ELISA (see below).

### 4.7. Immunofluorescence and Flow Cytometry Analysis

CD80/86 expression of MDDCs was assessed by staining with fluorochrome-conjugated antibodies or isotype controls (from B&D or eBiosciences, see above). Cells were then fixed with 1% PFA and analyzed by flow cytometry (using a Gallios flow cytometer by Beckman Coulter: 10 colors, three lasers). An analysis was performed by FlowJo10.0.7 (Tristar, Ashland, OR, USA).

### 4.8. ELISA for Cytokines/Chemokines in Cell Culture and Human Plasma

The plasma of SSc was diluted 1:4 in phosphate buffer solution (PBS) to assess the presence of IFN-α, TNF-α, and IL-6. Culture supernatants of MDDCs were diluted 1:2 or 1:3 or 1:4, depending on the cytokine analyzed. IFN-α was determined by ELISA (MabTech, Cincinnati, OH, USA, product code 3425-1H-6), as described [6]; TNF-α by ELISA (MabTech, Cincinnati, OH, USA, product code 3425-1H-6), IL-8 by MabTech (Cincinnati, OH, USA, product code 3560-1H-6); IL-6 by MabTech (Cincinnati, OH, USA, product code 3460-1H-6); pro-collagen by the Human Pro-collagen I Alpha 1 Duo Set >Elise Kit for 5 plates (product code DY6220-05; R&B Systems, Frankfurt, Germany) [6]; fibronectin by the Human Fibronectin DuoSet ELISA Kit for 5 plates (product code DY1918-05; R&B systems, Frankfurt, Germany). CXCL4 in plasma was tested by diluting plasma 1:100 in PBS using the Human CXCL4 DuoSet ELISA (R&D Systems).

### 4.9. In-House ELISA for the Determination of CXCL4-RNA Complexes

CXCL4–RNA complexes were identified using a capture ELISA. The capturing antibody, 2 μg per ml, of the mouse anti-human CXCL4 antibody (from Human CXCL4 DuoSet ELISA, R&D Systems) was coated to 96-well plates (100 μL) overnight at room temperature. After blocking in PBS 1% BSA (200 μL), plasma (100 μL diluted to 1:100 in 1% BSA in PBS) was added and incubated for 2 h at RT. After incubation, wells were washed three times with 200 μL of 0.05% Tween 20 in PBS, and the HRP-conjugated anti-RNA (anti-rRNA (Y10b): sc-33678, 200 ug/mL, Santa Cruz Biotechnology, Inc., Dallas, TX, USA) was added for 1 h at room temperature. After washing, the chromogenic substrate 3.3′, 5.5′-tetramethylbenzidine (TMB) was added and incubated in the dark; the absorbance was measured at 450 nm after stopping the reaction using 2 N HCl. Plasma samples were considered positive when the OD was above an established cut-off, which was calculated as the mean plus two times the standard deviation of the OD values obtained with HD plasma.

### 4.10. Visualization of Protein-Bound RNA by Western Blot

Plasma samples were solubilized with 1X RIPA buffer (in mM): 100 Tris-HCl, pH 7.5, 600 NaCl, 4% (*w*/*v*) Triton X-100, 4% (*v*/*v*) sodium deoxycholate, 0.4% sodium dodecyl sulfate (SDS) (*v*/*v*), 0.4 PMSF, and protease inhibitors (Complete). Then, they were incubated with 50% (*w*/*v*) protein G agarose beads (Pierce) for 3 h at 4 °C, clarified by centrifugation, and incubated overnight at 4 °C in a rotating wheel with a monoclonal anti-CXCL4 antibody ab9561 (Abcam, Cambridge, UK) (2 µg/sample) or IgG previously conjugated with 50% (*w*/*v*) Protein A agarose beads (Pierce). The immunoprecipitated immune complexes (IPs) were washed three times with RIPA buffer and two times with ice-cold Tris-buffered saline—Tween (150 mm NaCl, 1 mm dithiothreitol, 20 mm Tris-HCl, pH 7.5, 0.1% Tween 20, and protease inhibitors). After washing, the samples were subjected to SDS-PAGE, transferred to a nitrocellulose membrane, and immunoblotted with the polyclonal anti-CXCL4 antibody (Cat. No. ab9561, 100 µg- Abcam, Cambridge, UK) or with the monoclonal anti-RNA antibody (anti-rRNA Antibody (Y10b) 0.1 mL, Cat. No. NB 100-662, Novus Biologicals, Littleton, CO, USA). The immunoreactive bands were detected by chemiluminescence coupled to peroxidase activity (ECL Kit; Thermo Scientific, Cat. No. 32106) and quantified using a Bio-Rad ChemiDoc XRS system.

### 4.11. Statistical Analyses

Differences between mean values were assessed by Mann–Whitney test (one-tailed or two-tailed). Statistical significance was set at *p* ≤ 0.05. Correlation analyses were performed by Spearman’s rank correlation tests. Data were analyzed, and correlations were performed using GraphPad Prism 7.0.

## Figures and Tables

**Figure 1 ijms-24-00653-f001:**
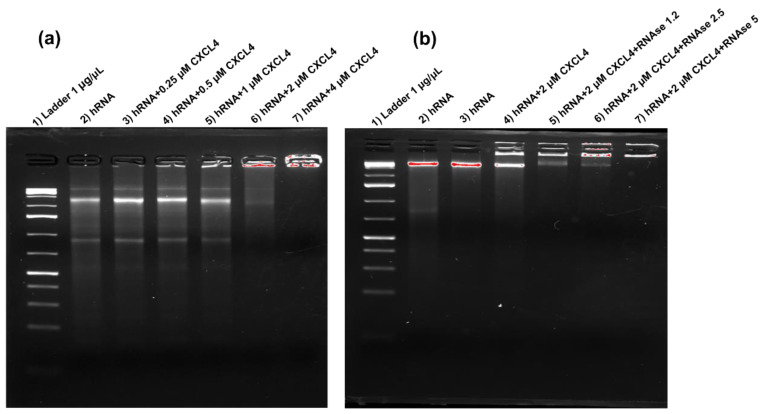
CXCL4 binds self-RNA in a dose-dependent manner and protects it from enzymatic degradation. (**a**) Human RNA at different concentrations was incubated for 30 min at 37 °C with different μMolar concentrations of CXCL4. The complexes were subsequently loaded onto 2% agarose and the RNA was colored with SYBR Green. Delayed RNA migration indicates the binding of CXCL4 to the RNA. (**b**) The RNA was incubated with CXCL4 in the presence or absence of different doses of RNAse. The DNA bound to CXCL4 is protected from degradation. Experiments in (**a**,**b**) were repeated 2 times.

**Figure 2 ijms-24-00653-f002:**
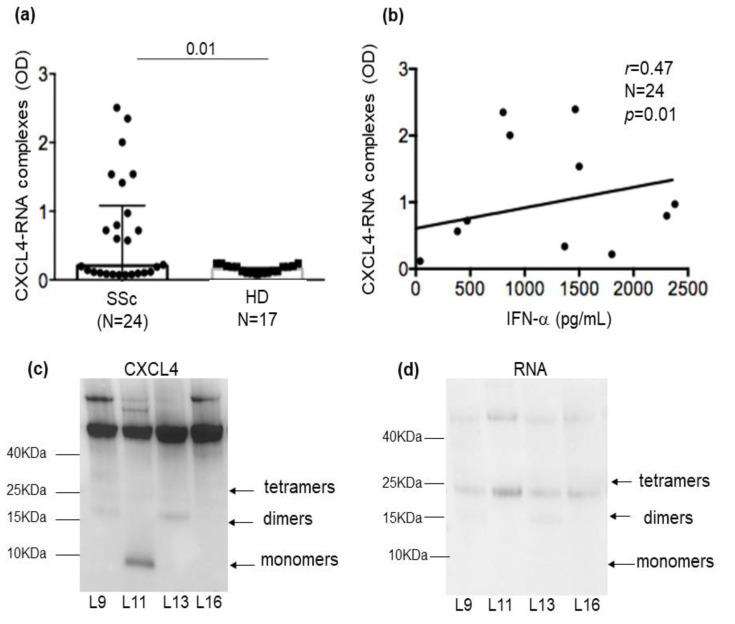
(**a**) CXCL4-RNA complexes are present in SSc plasma. Immune complexes of RNA with CXCL4 were determined with an in-house ELISA (see Methods) and results are expressed as optical density (OD) in SSc patients and HDs. Horizontal bars are the means; vertical bars are standard errors of the mean (SEM); *p*-values are from the Mann–Whitney test. (**b**) OD values were plotted against IFN-α concentrations measured in the SSc plasma. Correlation was calculated by Spearman’s correlation test. Coefficient of correlation r, significance *p,* and sample size N are indicated in the figure. (**c**) CXCL4 was immune-precipitated using an anti-CXCL4 antibody from the plasma of SSc patients, and the immune-precipitated material was separated by SDS-PAGE (10–20% Tricine gel) and transferred to nitrocellulose. The blot was probed with a monoclonal anti-CXCL4 antibody (**c**) and a polyclonal anti-RNA antibody (**d**) and revealed by ECL. The arrowheads on the right indicate bands corresponding to major CXCL4 forms. Molecular mass markers are indicated on the left.

**Figure 3 ijms-24-00653-f003:**
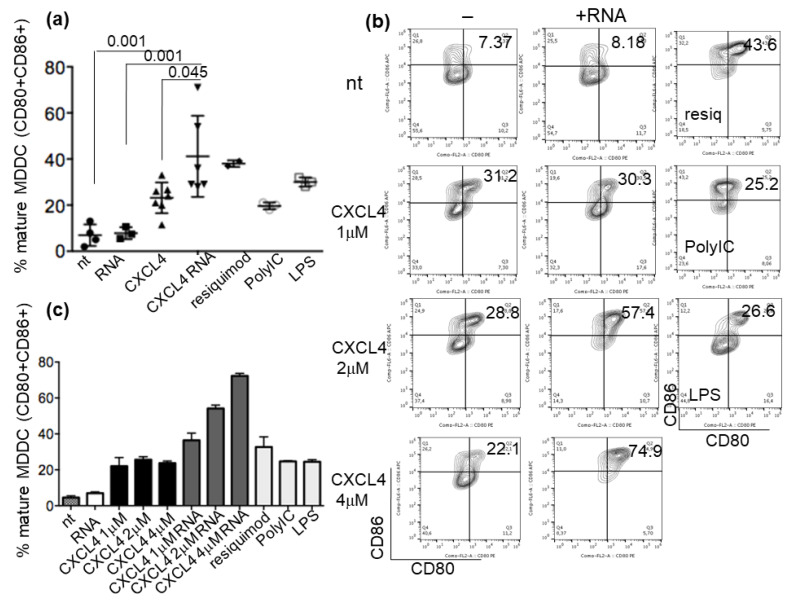
MDDCs are stimulated to maturity by CXCL4-RNA complexes. MDDCs were either untreated (nt) or treated with RNA alone or CXCL4 alone or with different amounts of CXCL4-RNA complexes (**b**,**c**) or with TLR agonists: resiquimod (TLR8), PolyIC (TLR3), or LPS (TLR4). MDDC maturation was evaluated by flow cytometry and is expressed as a percentage of double-positive (CD80^+^CD86^+^) cells. (**a**) Results from 3 to 5 experiments (depending on the stimulation) performed with the MDDCs of different donors. *p*-values were by Mann–Whitney’s test. (**b**) Flow cytometry representative plots of dose–response to different concentrations of immune complexes. In (**c**), the same representative experiment was visualized as histograms and standard errors (vertical bars), which were performed in triplicate.

**Figure 4 ijms-24-00653-f004:**
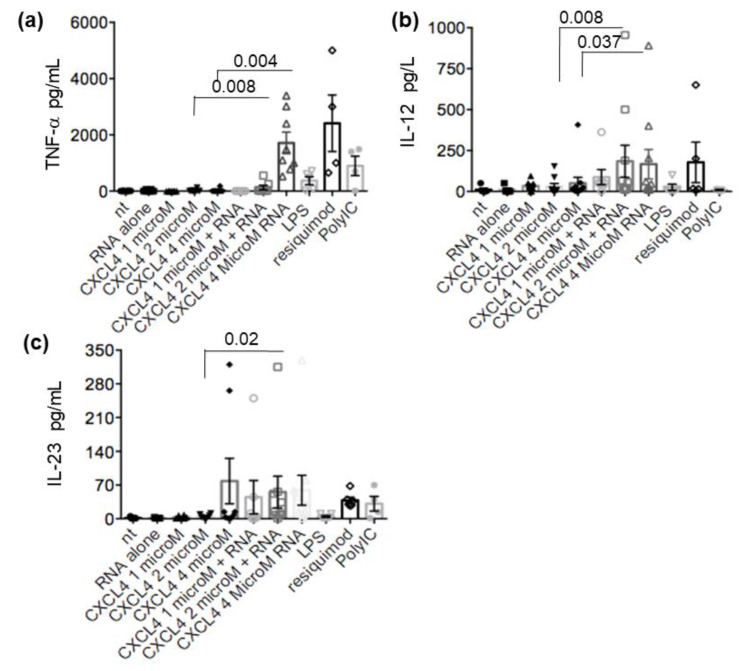
MDDCs are stimulated to produce pro-inflammatory cytokines by CXCL4-RNA complexes. MDDCs were either untreated (nt) or treated with RNA alone or CXCL4 alone or with different amounts of CXCL4-RNA complexes (**b**,**c**) or with TLR agonists. Production of cytokines, TNF-**α** (**a**), IL-12 (**b**), and IL-23 (**c**) was evaluated by ELISA on culture supernatants harvested after 48 h. (**a**–**c**) Results from 3 to 8 experiments (depending on the stimulation) performed with the MDDCs of different donors. *p*-values were by Mann–Whitney’s test.

**Figure 5 ijms-24-00653-f005:**
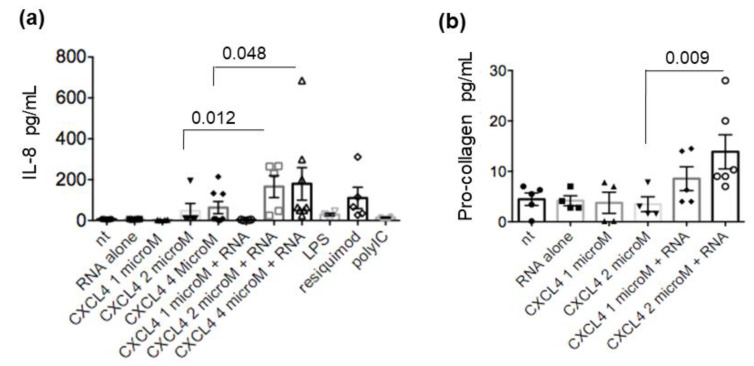
MDDCs are stimulated to produce pro-inflammatory/pro-fibrotic factors by CXCL4-RNA complexes. MDDCs were either untreated (nt) or treated with RNA alone or CXCL4 alone or with different amounts of CXCL4-RNA complexes or with TLR agonists. Production of the chemokine IL-8 (**a**) and pro-collagen (**b**) was evaluated by ELISA on culture supernatants harvested after 48 h. Results are from 3 to 8 experiments (depending on the stimulus) performed with MDDCs of different donors. *p*-values were by Mann–Whitney’s test.

**Figure 6 ijms-24-00653-f006:**
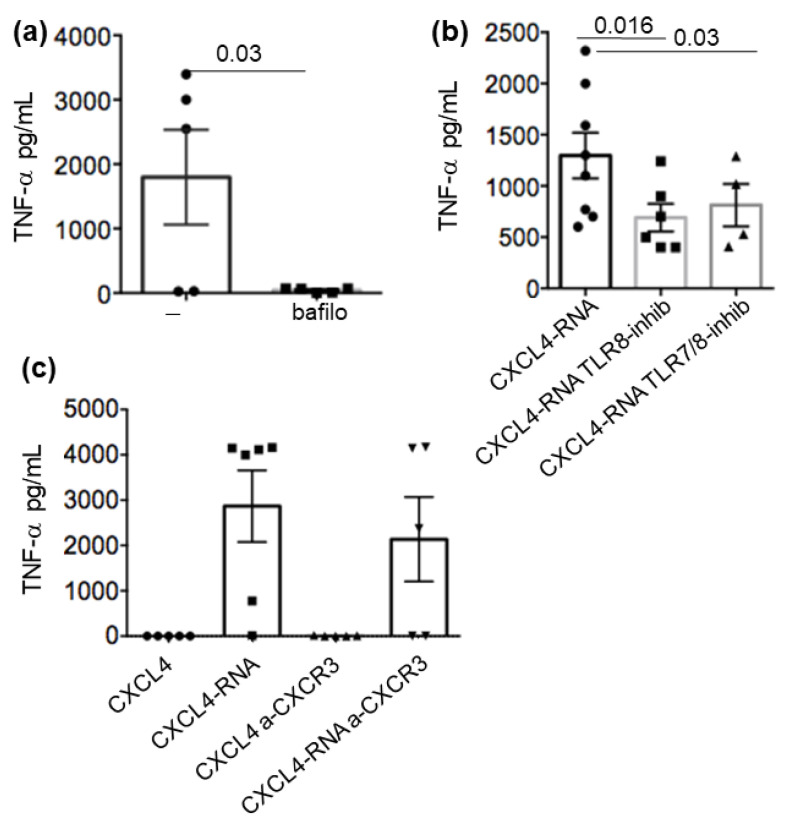
MDDC stimulation by CXCL4-RNA complexes is dependent on endosomal-sensing TLRs and independent of CXCR3. MDDCs were treated with CXCL4-RNA complexes in the presence of bafilomycin (**a**) or specific inhibitors of TLR7 and TLR8 (**b**) (see Methods) to block the stimulation of endosomal TLRs. Production of TNF-α was evaluated by ELISA, as above. (**c**) MDDCs were stimulated by CXCL4-RNA complexes in the presence of an anti-CXCR3 blocking antibody and TNF-α measured as in (**a**,**b**). Results from the experiments in all panels are from 3 to 9 experiments (depending on the stimulus) performed with the MDDCs of different donors. *p*-values were by Mann–Whitney’s test.

**Figure 7 ijms-24-00653-f007:**
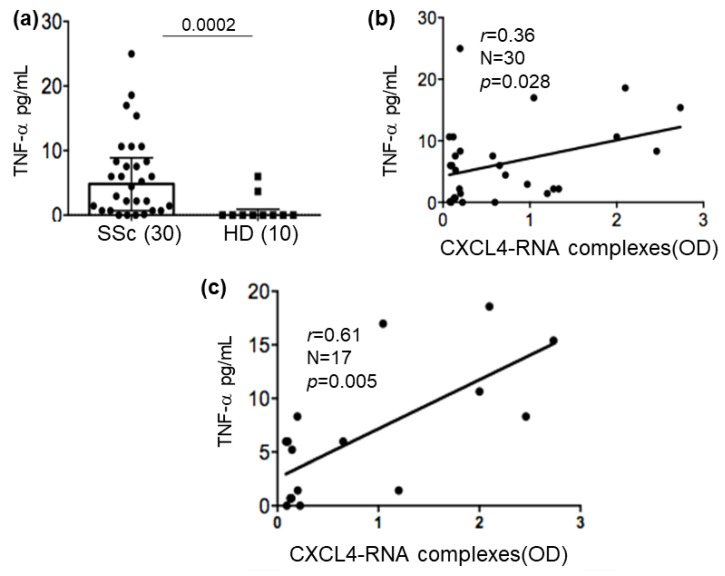
TNF-α in the plasma of SSc patients correlates with CXCL4-RNA complexes. Plasma concentrations of TNF-α were measured by ELISA in SSc and HD samples (**a**), and data on cytokine concentration were plotted against amounts of CXCL4-RNA complexes in the same plasma, expressed as OD (**b**). In (**c**), only SSc plasma samples positive for CXCL4-RNA complexes were selected. *p*-values in (**a**) are from the Mann–Whitney test; correlations in (**b**,**c**) were calculated by Spearman’s correlation test. Correlation coefficient r, significance *p*, and sample size N are reported on the graphs.

**Table 1 ijms-24-00653-t001:** Clinical characteristics of patients with SSc.

Characteristics ^1^	SSc (34)	HD (24)
Sex (M/F)	1/33	13/11
Age	52 (32–71)	50 (31–55)
Early	14	-
Long-lasting	20	-
RNA pol	6	-
ANA positivity (ATA/ACA)	29/2	-
PAH	20/14	-
Lung fibrosis (yes/no)	20/13	-
Prednisolone use	59%	-
Diff/lim	33/1	-

^1^ PAH: Pulmonary arterial hypertension; ANA: anti-nuclear antibodies (abs); ATA: anti-topoisomerase abs; ACA: anti-centromere abs; RNA pol: anti-RNA polymerase abs; Diff/lim: diffuse or limited SSc.

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
