# Peer review of "CXCL4-RNA Complexes Circulate in Systemic Sclerosis and Amplify Inflammatory/Pro-Fibrotic Responses by Myeloid Dendritic Cells"

_ijms, 2022, doi:10.3390/ijms24010653_

Round 1
Reviewer 1 Report
If CXCL4-DNA complexes elicit IFN-I production from pDCs, and IFN-I priming is a requirement for CXCL4-RNA induced mDC maturation, does it imply that both DNA and RNA complexes need to form to elicit pDC and mDC maturation and activation of pro-inflammatory cytokine production?
Author Response
We thank the reviewer for this question, which is important for clarifying the all picture.
Both types of nucleic acids are released with cell death, and they stimulate different TLRs. Since the myeloid DCs (in humans), which are of great importance in immunity, can only recognize RNA (using a TLR), we believe that both complexes are necessary. So a first response will be yes.
We know though, from the previous literature, included papers from Radstake’s group (that we have cited in the references), that CXCL4 can stimulate some functions of the human myeloid DC also independently from its binding capacity to RNA. Our paper points to underline that a stimulation of CXCL4-RNA complexes, especially in the presence of an IFN-I dominated environment, will amplify these maturation/activation/cytokine production and should be taken as a proof of a further role of CXCL4 in SSc and in the activation of immunity in general.
We added a few words on these aspects in the discussion, as the question could be rise by the readers of the paper.
Reviewer 2 Report
The authors show that CXCL4-RNA complexes circulate in SSc blood and correlate with both IFN-I and TNF-alfa. CXCL4, also increased the secretion by MDDC of fibronectin and IL-6. This work is interesting and significant to readers as TNF-α is a factor involved in SSc. It would be interesting to include patients with diffuse and limited cutaneous disease to determine the expression of CXCL-4-RNA in different clinical phenotypes.
Author Response
We thank the reviewer for the appreciation of our work.
Regarding the limited and diffuse SSc, yes, it is true that it is important to know whether the presence of CXCL4-RNA complexes can distinguish (or not) this two important disease subtypes. Unfortunately, our cohort from Policlinico Umberto I was mainly composed by diffuse SSc patients. So, we need to find a new cohort with more limited cutaneous disease patients. And to make this analysis we need a larger cohort. We planned to do this, and an ongoing project will analyze these complexes (and also CXCL4-DNA complexes in the two disease types and in early disease). For the moment, the composition of our cohort is a limitation that we have further highlighted in the Discussion.
Round 2
Reviewer 1 Report
I appreciate the addition to the Discussion that the authors have included in the manuscript. I have no further comments.
Author Response
We thank the reviewer.
Best regards
Loredana Frasca